# Illuminating the Genetic Basis of Congenital Heart Disease in Patients with Kabuki Syndrome

**DOI:** 10.3390/diagnostics14080846

**Published:** 2024-04-19

**Authors:** Chung-Lin Lee, Chih-Kuang Chuang, Ming-Ren Chen, Ju-Li Lin, Huei-Ching Chiu, Ya-Hui Chang, Yuan-Rong Tu, Yun-Ting Lo, Hsiang-Yu Lin, Shuan-Pei Lin

**Affiliations:** 1Department of Pediatrics, MacKay Memorial Hospital, Taipei 10449, Taiwan; clampcage@gmail.com (C.-L.L.); mingren44@gmail.com (M.-R.C.); g880a01@mmh.org.tw (H.-C.C.); wish1001026@gmail.com (Y.-H.C.); 2Institute of Clinical Medicine, National Yang-Ming Chiao-Tung University, Taipei 112304, Taiwan; 3Department of Rare Disease Center, MacKay Memorial Hospital, Taipei 10449, Taiwan; andy11tw.e347@mmh.org.tw; 4Department of Medicine, Mackay Medical College, New Taipei City 25245, Taiwan; 5Department of Nursing, Mackay Junior College of Medicine, Nursing and Management, Taipei 112021, Taiwan; 6Division of Genetics and Metabolism, Department of Medical Research, MacKay Memorial Hospital, Taipei 10449, Taiwan; mmhcck@gmail.com (C.-K.C.); likemaruko@hotmail.com (Y.-R.T.); 7College of Medicine, Fu-Jen Catholic University, Taipei 24205, Taiwan; 8Division of Endocrine & Medical Genetics, Department of Pediatrics, Chang Gung Children’s Medical Center, Chang Gung Memorial Hospital, Taoyuan 33378, Taiwan; jllin001@gmail.com; 9Department of Medical Research, China Medical University Hospital, China Medical University, Taichung 40402, Taiwan; 10Department of Infant and Child Care, National Taipei University of Nursing and Health Sciences, Taipei 11219, Taiwan

**Keywords:** congenital heart defect, Kabuki syndrome, *KMT2D* gene, Taiwan

## Abstract

Congenital heart defects (CHDs) affect a substantial proportion of patients with Kabuki syndrome. However, the prevalence and type of CHD and the genotype–phenotype correlations in Asian populations are not fully elucidated. This study performed a retrospective analysis of 23 Taiwanese patients with molecularly confirmed Kabuki syndrome. Twenty-two patients presented with pathogenic variants in the *KMT2D* gene. Comprehensive clinical assessments were performed. A literature review was conducted to summarize the spectrum of CHDs in patients with Kabuki syndrome. In total, 16 (73.9%) of 22 patients with pathogenic *KMT2D* variants had CHDs. The most common types of CHD were atrial septal defects (37.5%), ventricular septal defects (18.8%), coarctation of the aorta (18.8%), bicuspid aortic valve (12.5%), persistent left superior vena cava (12.5%), mitral valve prolapse (12.5%), mitral regurgitation (12.5%), and patent ductus arteriosus (12.5%). Other cardiac abnormalities were less common. Further, there were no clear genotype–phenotype correlations found. A literature review revealed similar patterns of CHDs, with a predominance of left-sided obstructive lesions and septal defects. In conclusion, the most common types of CHDs in Taiwanese patients with Kabuki syndrome who presented with *KMT2D* mutations are left-sided obstructive lesions and septal defects.

## 1. Introduction

Kabuki syndrome is a rare, genetically heterogeneous disorder first described by Kuroki et al. in 1981 and Niikawa et al. in the same year [1,2]. The condition is characterized by distinct facial features, intellectual disability, growth deficiency, skeletal anomalies, and various congenital malformations, including congenital heart defects (CHDs) [3,4,5]. The prevalence rate of CHDs in Kabuki syndrome varies widely. That is, it ranges from 28% to 80% based on clinical series prior to the identification of the causative genes [6,7,8,9,10,11,12]. The anatomical types of CHDs described in these early reports include left-sided obstructive lesions, septal defects, and conotruncal anomalies [6,9,10,12].

In 2010, pathogenic variants in the *KMT2D* gene (*MLL2*) were found to be the primary cause of Kabuki syndrome, accounting for 55–80% of all cases [13,14,15,16,17,18,19]. Subsequent studies have revealed that 9–14% of *KMT2D*-negative patients and <5% of patients with Kabuki syndrome presented with pathogenic variants in the *KDM6A* gene [20,21,22]. More recent studies have reported variants in the *RAP1A*, *RAP1B*, and *HNRNPK* genes in a small proportion of patients [23,24,25].

With the discovery of the molecular characteristics of Kabuki syndrome, the prevalence and types of CHDs in patients with confirmed genetic diagnoses can now be evaluated. Such studies can provide insights into the genotype–phenotype correlations and guidance on clinical management. The current study aimed to analyze the prevalence, anatomic types, and genetic characteristics of CHDs in 23 patients with Kabuki syndrome and the pathogenic variants in the *KMT2D* gene. Moreover, a literature review on CHDs in patients with molecularly confirmed Kabuki syndrome was performed.

## 2. Materials and Methods

### 2.1. Patient Cohort and Data Collection

We performed a retrospective study of 23 Taiwanese patients with molecularly confirmed Kabuki syndrome evaluated at MacKay Memorial Hospital between 2012 and 2023. Kabuki syndrome was diagnosed based on the presence of characteristic clinical features, as described in a previous study [3,4,5]. Comprehensive clinical assessments, including detailed physical examinations, were conducted by experienced medical geneticists. All patients underwent cardiac evaluations, including chest radiography, electrocardiography, and two-dimensional and color Doppler echocardiography.

#### Molecular Analysis

Genomic DNA was isolated from peripheral blood leukocytes using standard protocols. Targeted resequencing of the *KMT2D* gene was performed using a custom-designed Truseq Custom Amplicon panel and the MiSeq sequencing platform (Illumina, San Diego, CA, USA). The panel covered 33,255 base pairs with 135 amplicons, each with 500 base pairs, achieving 99% coverage of the target regions. Variant calling and visualization were performed using Illumina Variant Studio v3.0 and Integrative Genome Viewer v2.9.4, respectively. All identified variants were validated via Sanger sequencing. In silico prediction tools (Alamut Software v2.15) were used to assess the pathogenicity of novel missense variants. Parental testing was performed if available to determine whether the variants were de novo or inherited. Variants were classified as pathogenic if they resulted in protein truncation, altered splicing, or missense changes proven to be de novo in at least one patient. If parental samples were unavailable, missense variants were considered pathogenic if they were identified in multiple patients or met the ACMG criteria for pathogenicity. Variants of uncertain significance and those inherited from an unaffected parent were excluded from this study. All patients in our cohort were diagnosed before 2013. Thus, genetic testing was limited to the *KMT2D* gene and did not include other genes subsequently associated with Kabuki syndrome.

### 2.2. Literature Review

A comprehensive literature review was conducted to summarize the prevalence and types of CHDs reported in previous cohorts of patients with molecularly confirmed Kabuki syndrome. The review followed the Preferred Reporting Items for Systematic Reviews and Meta-Analyses (PRISMA) guidelines [26]. Electronic databases, including PubMed, Embase, and Web of Science, were searched for relevant articles using the following search terms: “Kabuki syndrome”, “congenital heart defect”, “cardiac malformation”, “*KMT2D*”, “*KDM6A*”, “*RAP1A*”, “*RAP1B*”, and “*HNRNPK*”. The search was limited to articles published in English up to 29 February 2024. The inclusion criteria were as follows: (1) studies reporting patients with molecularly confirmed Kabuki syndrome, (2) presence of detailed cardiac phenotypic information, and (3) original research articles. Case reports, review articles, and studies without molecular confirmation of Kabuki syndrome diagnosis were excluded. The titles and abstracts of the retrieved articles were screened for relevance, and the full texts of the selected articles were subsequently reviewed. The reference lists of the included articles were also manually searched for additional relevant studies. The data extracted from the eligible studies included the number of patients with molecularly confirmed Kabuki syndrome, the prevalence of CHDs, and the types of CHDs reported. A PRISMA flow diagram depicting the literature search and study selection process is provided in Figure 1.

### 2.3. Statistical Analysis

Descriptive statistical analyses were performed to summarize the prevalence and distribution of CHDs in our cohort and in the literature. Categorical variables were expressed as frequencies and percentages, and continuous variables were reported as means and standard deviations or medians and ranges, as appropriate.

## 3. Results

In total, 22 (95.7%) of 23 patients with Kabuki syndrome presented with pathogenic variants on molecular analysis of the *KMT2D* gene (Table 1). The identified variants included 14 frameshift mutations, 6 nonsense mutations, 1 missense mutation, and 1 splice site mutation. Figure 2 shows the distribution of *KMT2D* variants. c.15461G>A was the most common variant (*n* = 4, 14%), followed by c.7144C>T (*n* = 10%) and other variants with lower frequencies.

Sixteen (73.9%) patients had congenital heart defects. Figure 3 shows the distribution of congenital heart defects. Table 2 depicts the prevalence of different anatomic types of congenital heart defects in patients with *KMT2D* pathogenic variants and cardiac malformation. ASD (*n* = 6, 37.5%) was the most common anomaly, followed by ventricular septal defect (VSD) (*n* = 3, 18.8%), coarctation of the aorta (CoA) (*n* = 3, 18.8%), bicuspid aortic valve (*n* = 2, 12.5%), persistent left superior vena cava (*n* = 2, 12.5%), mitral valve prolapse (*n* = 2, 12.5%), mitral regurgitation (*n* = 2, 12.5%), and patent ductus arteriosus (*n* = 2, 12.5%). One (6.3%) patient presented with other cardiac abnormalities, including mitral stenosis, aberrant right subclavian artery, vascular ring, dilated aortic root, interrupted aortic arch, subaortic ridge, left isomerism of the heart, aortic dissection, redundant mitral valve, redundant tricuspid valve, and thick interventricular septum.

Patients with the same variant (e.g., c.15461G>A, c.11944C>T) exhibited variable cardiac phenotypes. Hence, no clear genotype–phenotype correlations were observed. Interestingly, patient 23, the father of patient 8, also harbored a pathogenic *KMT2D* variant (c.10741-7A>G), indicating possible vertical transmission of the variant in this family.

## 4. Discussion

Our study found that approximately 69.6% of Taiwanese patients with molecularly confirmed Kabuki syndrome caused by pathogenic *KMT2D* variants presented with CHDs. This prevalence is consistent with the higher end of the range reported in previous clinical series prior to the identification of the causative genes (28–80%) [6,7,8,9,10,11,12]. The most common types of CHDs in our cohort were left-sided obstructive lesions (particularly coarctation of the aorta and bicuspid aortic valve), septal defects (atrial and ventricular), and persistent left superior vena cava. This distribution is in accordance with the spectrum of defects described in earlier reports [6,9,10,12].

Aortic coarctation and bicuspid aortic valve were the most frequent left-sided obstructive lesions in our study, which is consistent with their predominance noted in prior series [7,12,27]. Interestingly, coarctation was found to often co-occur with additional left-sided anomalies such as mitral valve abnormalities, resembling the constellation of defects observed in Shone complex [28]. This association has been reported in other Kabuki syndrome cohorts [27,29]. Thus, this pattern of multiple left-sided obstructions may be a hallmark of the condition.

Septal defects were the second most common category of CHDs in our patients. The prevalence (32%) rate of septal defects in this study was lower than that in previous studies, which reported rates as high as 50–60% [19,27]. This discrepancy may be attributed to differences in the comprehensiveness of cardiac evaluations and the inclusion of patent foramen ovale as a septal defect in some series.

CHDs such as interrupted aortic arch, subaortic stenosis, and vascular rings have been rarely observed in our cohort. However, these conditions are common in other studies [27,30,31,32]. Hence, the possible cardiac manifestations of Kabuki syndrome significantly vary.

There were no clear genotype–phenotype correlations with respect to the type or severity of CHDs associated with specific *KMT2D* variants. The most common variants in our cohort (c.15461G>A and c.7144C>T) were associated with variable cardiac phenotypes, ranging from normal hearts to complex defects. This lack of correlation is consistent with the findings of previous studies [14,18,19]. Further, it indicates that other genetic, epigenetic, or environmental factors may influence the cardiac outcomes of patients with *KMT2D* mutations.

Interestingly, we observed a case of possible vertical transmission of a pathogenic *KMT2D* variant (c.10741-7A>G) from an affected father to his son, both of whom had Kabuki syndrome but differed in terms of cardiac phenotypes (normal heart in the father, patent ductus arteriosus in the son). In most cases, Kabuki syndrome is sporadic. However, some studies have reported rare instances of familial transmission [33,34,35]. This underscores the importance of parental testing and genetic counseling for families affected by Kabuki syndrome.

Based on a review of the literature, patients with *KDM6A* variants had a lower frequency of CHDs (approximately 45%) than those with *KMT2D* mutations [20,21,22,36]. The anatomic types of CHDs associated with *KDM6A* variants included septal defects, hypoplastic right ventricle, pulmonary stenosis, and aortic abnormalities (e.g., abnormal shape, subaortic membrane, bicuspid valve, and coarctation) [20,22,36,37]. Notably, right-sided lesions can be more prevalent in patients with *KDM6A* (KS2) than in those with *KMT2D* (KS1) [22].

Recent studies have identified additional genes associated with Kabuki syndrome, including *RAP1A*, *RAP1B* [23], and *HNRNPK* [24,25]. *RAP1A* and *RAP1B* dysfunction results in aberrant MEK/ERK signaling, indicating an overlap in the pathophysiology of Kabuki syndrome caused by these genes and the RASopathies [23]. A patient with a *RAP1B* variant presented with CHD. However, the specific type was not identified [23]. Patients with *HNRNPK* variants exhibited heterogeneous CHDs, including septal defects, bicuspid aortic valve with aortic root dilatation, and atrioventricular septal defect [24,25]. Interestingly, a patient with a deletion encompassing *HNRNPK* presented with atrioventricular canal defect [38]. Thus, there can be a possible specific association between this gene and the development of this CHD type.

Experimental models have provided valuable insights into the role of Kabuki syndrome genes in cardiac development. Lee et al. [39] showed that female *KDM6A* knockout mice exhibited severe cardiac malformations, and heterozygous mice had normal cardiac morphogenesis. Van Laarhoven et al. [40] investigated the effects of kmt2d, kdm6a, and kdm6al knockdown on cardiac development in zebrafish, thereby observing morphological defects in all three morphant groups, with the most pronounced abnormalities in kmt2d morphants. These defects included abnormal atrial and ventricular development, myocardial wall bulging, and impaired cardiac looping [40]. Ang et al. [41] conducted further studies on mice. The results showed the essential role of kmt2d in regulating cardiac gene expression during heart development, primarily via H3K4 di-methylation. Haploinsufficient mice did not exhibit gross cardiac morphological differences. Nevertheless, they presented with a narrowing of the ascending aorta and increased aortic valve peak velocity [41].

These findings from experimental models support the important roles of *KMT2D* and *KDM6A* in cardiac development and provide mechanistic insights into the CHD pathogenesis in Kabuki syndrome. More genes associated with Kabuki syndrome are identified. However, further studies must be conducted to elucidate their specific contributions to cardiac phenotypes and explore potential genotype–phenotype correlations. Recent studies have provided new insights into the potential molecular mechanisms underlying the congenital heart defects in Kabuki syndrome. Serrano et al. [42] demonstrated that the KMT2D protein, which is deficient in Kabuki syndrome, directly interacts with and regulates the expression of NOTCH1, a key component of the NOTCH signaling pathway. The NOTCH pathway plays a crucial role in cardiac development, and perturbations in this pathway have been associated with various congenital heart defects [43]. The findings of Serrano et al. suggest that dysregulation of NOTCH signaling may contribute to the cardiac phenotypes observed in patients with Kabuki syndrome. This study provides a novel link between *KMT2D* function and NOTCH signaling, opening up new avenues for understanding the pathogenesis of congenital heart defects in Kabuki syndrome and potentially identifying new therapeutic targets. Further research is needed to elucidate the precise mechanisms by which *KMT2D* and other Kabuki syndrome genes interact with the NOTCH pathway and other signaling cascades to regulate cardiac development and to determine the extent to which these interactions contribute to the diverse cardiac phenotypes observed in patients with this disorder.

Previous studies have provided valuable insights into how the genetic variants identified in our patients with Kabuki syndrome contribute to the pathogenesis of specific cardiac malformations. In our study, the most common *KMT2D* variants were c.15461G>A and c.7144C>T. Ang et al. [41] found that the c.15461G>A variant results in a loss of function of the KMT2D protein, affecting the expression of key genes involved in cardiac development, particularly those involved in cardiac mesoderm development and ventricular formation. This may explain the cardiac phenotypes observed in patients harboring this variant, such as left-sided obstructive lesions and ventricular septal defects.

Similarly, Shpargel et al. [44] demonstrated that the c.7144C>T variant disrupts the interaction of KMT2D with components of the NOTCH1 signaling pathway, which plays a crucial role in regulating cardiac valve and endocardial cushion development. This may be related to the bicuspid aortic valve and cardiac valve abnormalities observed in patients with the c.7144C>T variant.

Furthermore, we identified other rare *KMT2D* variants, such as c.10741-7A>G and c.1113-40_1152del80insTA. Although the functional impact of these variants has not been extensively studied, a recent study suggested that similar intronic variants disrupt *KMT2D* function by interfering with its proper splicing [45]. This highlights the importance of further investigating the effects of these variants on cardiac development.

Collectively, these studies underscore the critical role of *KMT2D* variants in the pathogenesis of cardiac malformations in patients with Kabuki syndrome and provide molecular insights linking specific genetic variants to distinct cardiac phenotypes. As our understanding of the genetic basis of Kabuki syndrome and its impact on cardiac development continues to grow, we will be better equipped to understand and predict the clinical manifestations associated with specific variants and develop personalized patient management strategies.

The current study had several limitations. First, our cohort size was relatively small, reflecting the rarity of Kabuki syndrome. Hence, larger, multicenter studies must be conducted to better characterize the full spectrum and prevalence of CHDs in this population. Second, most patients were diagnosed before these genes were identified. Thus, a comprehensive screening of other genes associated with Kabuki syndrome (e.g., *KDM6A*, *RAP1A*, *RAP1B*, and *HNRNPK*) was not performed. Future studies should be performed to investigate the cardiac phenotypes associated with these genes to facilitate more comprehensive genotype–phenotype analyses.

Despite these limitations, our study provides important insights into the prevalence and types of CHDs in Taiwanese patients with Kabuki syndrome and *KMT2D* mutations. The high prevalence of left-sided obstructive lesions, particularly aortic coarctation associated with mitral abnormalities, emphasizes the need for comprehensive cardiac evaluations and long-term surveillance in this population. We recommend echocardiographic screening upon diagnosis to detect left-sided anomalies, and periodic follow-up to monitor for potential complications such as aortic dilatation [26,46].

The absence of patients with hypoplastic left heart syndrome (HLHS) in our cohort may be attributed to a potential survival bias, as our study population consisted of patients who were diagnosed with Kabuki syndrome and underwent genetic testing at our institution. This bias may have led to an underestimation of the frequency of more severe or life-threatening congenital heart defects, such as HLHS, in patients with Kabuki syndrome. Previous studies have reported the occurrence of HLHS in patients with Kabuki syndrome, suggesting that it is a part of the spectrum of congenital heart defects associated with this disorder [31,47]. However, the true prevalence of HLHS in Kabuki syndrome remains unknown due to the limited number of reported cases and the potential for survival bias in existing studies. Further research with larger, multi-center cohorts is necessary to better characterize the full range of congenital heart defects, including HLHS, in patients with Kabuki syndrome and to minimize the impact of potential survival bias on the reported frequencies of these defects.

## 5. Conclusions

CHDs are common and diverse in Taiwanese patients with Kabuki syndrome and *KMT2D* mutations, with left-sided obstructive lesions being the most prevalent. There were no clear genotype–phenotype correlations identified. Hence, further research with larger, multi-ethnic cohorts should be conducted to elucidate the genetic basis of CHDs in Kabuki syndrome and to provide individualized management strategies. Our findings underscore the importance of comprehensive cardiac evaluations and long-term surveillance in this population to optimize treatment outcomes and quality of life.

## Figures and Tables

**Figure 1 diagnostics-14-00846-f001:**
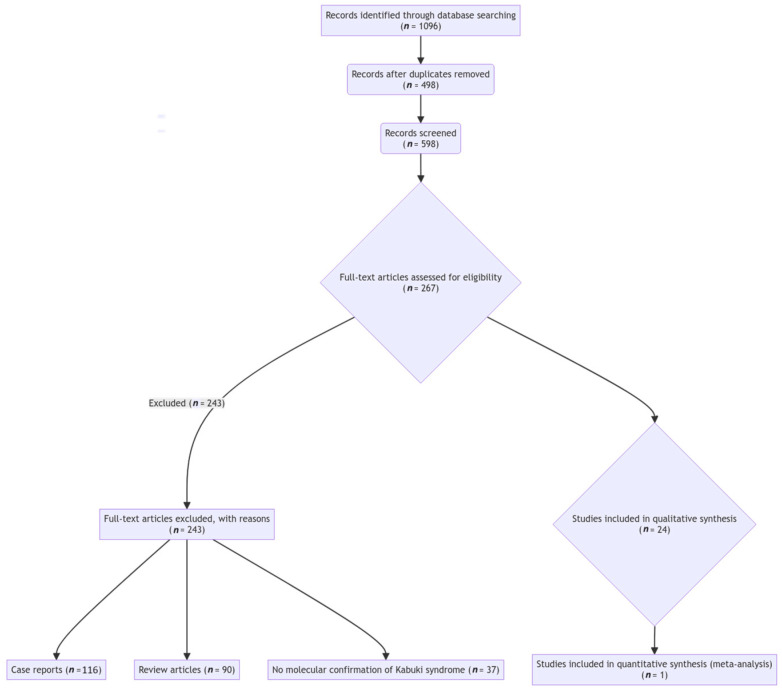
PRISMA flow diagram depicting the literature search and study selection process for the review of congenital heart defects in patients with molecularly confirmed Kabuki syndrome.

**Figure 2 diagnostics-14-00846-f002:**
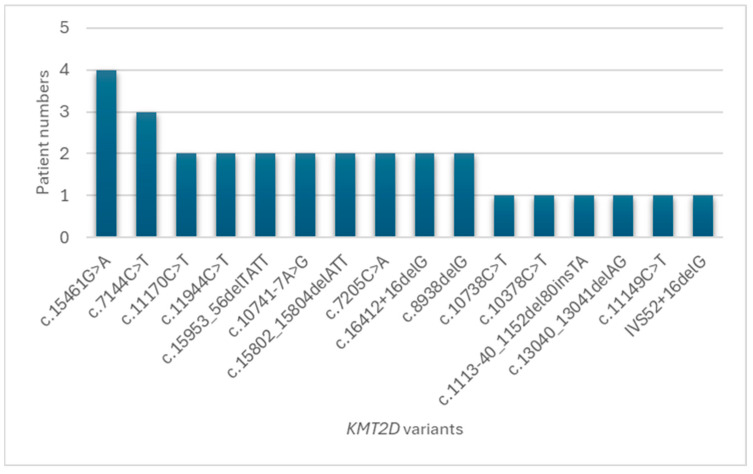
Prevalence of *KMT2D* variants in Taiwanese patients with Kabuki syndrome. The barchart illustrates the distribution of *KMT2D* variants in the study cohort. The c.15461G>A variant (14%) was the most prevalent, followed by c.7144C>T (10%), c.11170C>T (7%), and other variants (with lower frequencies). The percentages represent the proportion of patients harboring each specific *KMT2D* variant. The others category includes variants found in only one patient each.

**Figure 3 diagnostics-14-00846-f003:**
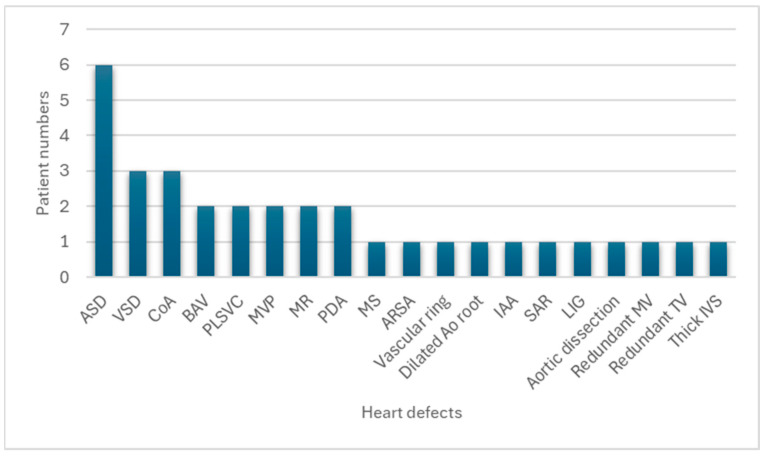
Distribution of congenital heart defects in Taiwanese patients with Kabuki syndrome. The bar graph displays the frequency of various congenital heart defects in the study cohort. ASD was the most common anomaly, followed by VSD, CoA, and other less frequent cardiac abnormalities. The numbers above each bar represent the number of patients affected by each specific congenital heart defect. MS, mitral stenosis; PLSVC, persistent left superior vena cava; MVP, mitral valve prolapse; MR, mitral regurgitation; BAV, bicuspid aortic valve; CoA, coarctation of the aorta; ARSA, aberrant right subclavian artery; ASD, atrial septal defect; VSD, ventricular septal defect; PDA, patent ductus arteriosus; IAA, interrupted aortic arch; SAR, subaortic ridge; LIG, left isomerism of the heart; IVS, interventricular septum; TV, tricuspid valve; MV, mitral valve.

**Table 1 diagnostics-14-00846-t001:** Molecular defects and congenital heart defects identified in 23 Taiwanese patients with Kabuki syndrome. The table includes data on sex, *KMT2D* cDNA and amino acid changes, and the specific types of congenital heart defects in each patient. Patient 23 is the father of patient 8. * Asterisk indicates a stop codon (also known as a termination codon or nonsense codon), which signals the end of the protein-coding sequence. Dash indicates that no amino acid change is associated with the corresponding nucleotide change, which may occur in cases of synonymous variants or variants affecting non-coding regions.

Patient	Sex	KMT2D cDNA Change	*KMT2D* Amino Acid Change	ACMG Classification	Congenital Heart Defect
1	M	c.15461G>A	p.Arg5154Gln	Likely Pathogenic	MS, PLSVC, MVP, MR, BAV, CoA, ARSA
2	M	c.10738C>T	p.Gln3580*	Pathogenic	ASD
3	M	c.10378C>T	p.Gln3460*	Pathogenic	VSD
4	M	c.11944C>T	p.Arg3982*	Pathogenic	VSD, redundant TV
5	M	c.1113-40_1152del80insTA	-	Likely Pathogenic	Vascular ring, PDA
6	F	c.15953_56delTATT	p.Leu5318Serfs*14	Pathogenic	ASD
7	F	c.13040_13041delAG,	p.Gln4347Argfs*24,	Pathogenic,	ASD
		c.7144C>T	p.Pro2382Ser	Uncertain Significance	
8	M	c.10741-7A>G	-	Likely Pathogenic,	PDA
		c.15802_15804delATT	p.Ile5268del	Pathogenic	
9	F	c.7205C>A	p.Ser2402*	Pathogenic	IAA, VSD, PLSVC, SAR, MVP, MR
10	F	c.15802_15804delATT	p.Ile5268del	Pathogenic	Normal
11	M	c.7205C>A	p.Ser2402*	Pathogenic	BAV
12	F	c.15953_56delTATT	p.Leu5318Serfs*14	Pathogenic	Normal
13	M	c.15461G>A	p.Arg5154Gln	Likely Pathogenic	Normal
14	M	c.11149C>T	p.Gln3717*	Likely Pathogenic	Normal
15	F	c.7144C>T	p.Pro2382Ser	Uncertain Significance	CoA, PDA, LIG, aortic dissection
16	F	c.7144C>T, c.16412 + 16delG	p.Pro2382Ser, -	Uncertain Significance, Likely Pathogenic	CoA
17	F	c.15461G>A	p.Arg5154Gln	Likely Pathogenic	Normal
18	M	c.8938delG	p.Ala2980Profs*24	Pathogenic	BAV
19	M	c.11170C>T, c.16412 + 16delG	p.Pro2382Ser, -	Uncertain Significance, Likely Pathogenic	redundant MV, redundant TV, ASD, thick IVS
20	F	c.8938delG	p.Ala2980Profs*24	Pathogenic	BAV
21	F	c.15461G>A	p.Arg5154Gln	Likely Pathogenic	Normal
22	F	IVS52 + 16delG	-	Likely Pathogenic	ASD
23	M	c.10741-7A>G	-	Likely Pathogenic	Normal

MS, mitral stenosis; PLSVC, persistent left superior vena cava; MVP, mitral valve prolapse; MR, mitral regurgitation; BAV, bicuspid aortic valve; CoA, coarctation of the aorta; ARSA, aberrant right subclavian artery; ASD, atrial septal defect; VSD, ventricular septal defect; PDA, patent ductus arteriosus; IAA, interrupted aortic arch; SAR, subaortic ridge; LIG, left isomerism of the heart; IVS, interventricular septum; TV, tricuspid valve; MV, mitral valve.

**Table 2 diagnostics-14-00846-t002:** Prevalence of different anatomic types of congenital heart defects in patients with *KMT2D* pathogenic variants and cardiac malformation in the current series.

Congenital Heart Defect	Present Series (Number of Patients with Congenital Heart Defect)	Present Series (%)
ASD, ostium secundum	6/16	37.5
VSD, perimembranous subaortic	3/16	18.8
Aortic coarctation	3/16	18.8
Bicuspid aortic valve	2/16	12.5
Persistent left superior vena cava	2/16	12.5
Mitral valve prolapse	2/16	12.5
Mitral regurgitation	2/16	12.5
Patent ductus arteriosus	2/16	12.5
Mitral stenosis	1/16	6.3
Aberrant right subclavian artery	1/16	6.3
Vascular ring	1/16	6.3
Dilated aortic root	1/16	6.3
Interrupted aortic arch	1/16	6.3
Subaortic ridge	1/16	6.3
Left isomerism of the heart	1/16	6.3
Aortic dissection	1/16	6.3
Redundant mitral valve	1/16	6.3
Redundant tricuspid valve	1/16	6.3
Thick interventricular septum	1/16	6.3

ASD, atrial septal defect; VSD, ventricular septal defect.

## Data Availability

The study’s original findings and the data supporting the reported conclusions are fully presented in the manuscript. Interested readers may contact the corresponding authors with any additional questions.

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
