# Peer review of "Illuminating the Genetic Basis of Congenital Heart Disease in Patients with Kabuki Syndrome"

_diagnostics, 2024, doi:10.3390/diagnostics14080846_

Round 1

Reviewer 1 Report

Comments and Suggestions for Authors

Overall this is a very nicely written retrospective review of a cohort of patients with Kabuki syndrome (gene-proven in all but one case).  Below are some specific comments/questions that I have:

1).  In the text of Figure 2, "POP" is mentioned.  What is that?

2).  I would not consider a PFO to be a congenital heart defect.  It occurs in about 20% of the general population (8% in this cohort).

3).  Can the authors comment on why there were not any patients with hypoplastic left heart syndrome in their cohort?  I assume there was a patient selection bias, specifically for patients that survive long-term.  This would be worth mentioning, i.e. they are likely underestimating the frequency of more complex CHDs in Kabuki syndrome.

4).  The authors should cite one recent study which suggested a role for NOTCH signaling in Kabuki syndrome (Serrano, et al.  PLOS Biol, 2019 Sept 3;17(9).

Reviewer 2 Report

Comments and Suggestions for Authors

This brief report by Lee et al studies 23 taiwanese patients with a confirmed molecular diagnosis of Kabuki syndrome. 

Literature review methodology needs to be expanded. A Prisma diagram would be helpful. Information about what databases were searched, what search terms were employed, how many articles the search returned, how many were included in the analysis, what were the inclusion and exclusion criteria should be added.

The Discussion paragraph could be expanded. What do previous studies say about how the genetic variants you have discovered in your patients are involved in the etiology of the respective heart malformations? Can you offer more molecular insights?

Round 2

Reviewer 2 Report

Comments and Suggestions for Authors

The authors have addressed my comments.